# TM9SF2 Maintains Golgi Integrity and Regulates Ricin-Induced Cytotoxicity

**DOI:** 10.3390/toxins17050218

**Published:** 2025-04-26

**Authors:** Yue Meng, Hongzhi Wan, Xinyu Wang, Lina Zhang, Ruozheng Xin, Lingyu Li, Yuhui Wang, Chengwang Xu, Hui Peng, Lu Sun, Bo Wang, Xiaotao Duan

**Affiliations:** 1School of Pharmacy, Qingdao University, Qingdao 266071, China; 2National Key Laboratory of Special Drugs for National Security, Academy of Military Medical, Beijing 100850, China; 3Military Medical Sciences Academy, Tianjin 300050, China; 4School of Pharmacy, Nanjing University of Chinese Medicine, Nanjing 210023, China; 5School of Pharmacy, Shenyang Pharmaceutical University, Shenyang 110016, China

**Keywords:** type II ribosome-inactivating toxin, TM9SF2, Golgi fragmentation, intercellular cholesterol transport

## Abstract

TM9SF2 belongs to a family of highly conserved nonaspanin proteins, and has been frequently identified as one of the important host factors for a plethora of lethal pathogens and toxins in previous genome-wide screening studies. We reported herein a novel molecular mechanism of TM9SF2 in mediating the cytotoxicity of ricin, a type II ribosome-inactivating protein. We first showed that TM9SF2 displays a non-redundant requirement for ricin-induced cytotoxicity within the nonaspanin family. Then we found that genetic interference of TM9SF2 substantially affects/remodels intracellular cholesterol trafficking, which results in abnormal cholesterol accumulation in Golgi compartments and causes severe Golgi fragmentation. The disruption of Golgi integrity and network impedes the retrograde transport of ricin and thus attenuates ricin-induced cytotoxicity. We further verified this mechanism by pharmacological manipulation of cholesterol metabolism (e.g., by using A939572 and avasimibe, etc.), which well restores the integrity of the Golgi apparatus and reverses the ricin-resistant phenotype induced by TM9SF2 knockdown. Our finding provides new mechanistic insights into the pathology and toxicology of ricin and could potentially be applied to other ribosome-inactivating toxins.

## 1. Introduction

Ricin is a type II ribosome-inactivating protein isolated from the seeds of the castor oil plant (*Ricinus communis*) [1,2]. Ricin poisoning can manifest in multiple symptoms, including vomiting, gastroenteritis, fever, and diarrhea, etc. [3,4,5]. The lethal dose (LD_50_) of ricin in humans differs significantly depending on the exposure route; the median lethal dose (LD50) for inhalation exposure is 3–5 μg/kg, while the oral LD50 is around 20 mg/kg [6,7].

Structurally, ricin consists of two polypeptide chains linked by disulfide bonds: an A chain (RTA) acts as an N-glycosidase, and a B chain (RTB) functions as the receptor-binding domain [8]. After internalization into the host cell, ricin follows a retrograde transport route from early endosomes via the Golgi apparatus to the endoplasmic reticulum (ER). At the ER, the two chains of ricin are reductively separated, and the free RTA is then translocated to the cytosol and inhibits protein synthesis by inactivating ribosomes [7,9,10].

TM9SF2 is a highly conserved member of the transmembrane 9 superfamily (which consists of TM9SF1-4). It bears a large N-terminal extracellular domain and nine transmembrane domains in its structure [11,12], and is predominantly localized in the early endosomes and the Golgi apparatus [13,14,15]. Functionally, TM9SF2 is linked to glycolipid biosynthesis and frequently shows upregulation in the cancerous context [12,16]. It has been reported that TM9SF2 mediates the cellular toxic effects of Shiga toxin (Stx). TM9SF2 silencing results in a decrease in the expression level of Gb3, the binding receptor of Stx on the cell surface, and thereby confers resistance to Stx toxicity [8,11,14]. A number of genome-wide screening studies have identified that TM9SF2 is one of the important host factors that may mediate the cellular toxic effects of Shiga toxin (Stx). It has been described that TM9SF2 silencing results in a decrease of the expression level of Gb3, the binding receptor of Stx on the cell surface, and thereby confers resistance to Stx toxicity [8,11,14]. Additionally, TM9SF2 depletion can lead to abnormal endosomal function, which is related to the internalization stage of Stx [8].

In this study, we verified that TM9SF2 is a critical host factor for mediating ricin-induced cytotoxicity in cellular models. We found that genetic interference of TM9SF2 significantly increases the cellular resistance to ricin exposure. Mechanistically, we demonstrated that TM9SF2 knockdown causes cholesterol accumulation at the Golgi and disrupts Golgi integrity, which impedes the retrograde transport of ricin and attenuates ricin-induced cytotoxicity. Pharmacological manipulation of cholesterol metabolism readily restores the Golgi network and reverses the ricin-resistant phenotype induced by TM9SF2 knockdown.

## 2. Results

### 2.1. TM9SF2 Displays Non-Redundant Requirement for Ricin-Induced Cytotoxicity Within the TM9SF Family

To verify the biological significance of TM9SF2 in ricin-induced cytotoxicity, we constructed three small different interfering RNAs (siRNAs, targeting non-overlapping parts of TM9SF2 mRNA) and performed genetic knockdown of endogenous TM9SF2 in HeLa cells. We observed that silencing of TM9SF2 dramatically enhanced cellular resistance to ricin-induced cytotoxicity, as demonstrated by increased CC_50_ values (siNC: 0.08 ng/mL; siTM9SF2 #1: 0.29 ng/mL; #2: 0.21 ng/mL; #3: 0.21 ng/mL) in MTT-based viability assays (Figure 1A). The knockdown efficiency was tested by Western blot analysis (Figure 1B). Crystal violet staining showed consistent results (Figure 1C). We further examined whether other members of the TM9SF family (TM9SF1, TM9SF3, and TM9SF4) are also involved in ricin-induced cytotoxicity. We did not observe any appreciable effects of these sibling proteins with regard to the cell viability upon ricin treatment (Figure 1D–F and Appendix A). These results demonstrated that TM9SF2 plays an indispensable and non-redundant role in ricin-induced cytotoxicity.

### 2.2. TM9SF2 Deficiency Impairs Cholesterol Translocation and Causes Fragmentation of Golgi

We next examined the cellular localization of TM9SF2 following 36-h exogenous transfection. The immunofluorescence results showed that TM9SF2 protein primarily resides in the Golgi apparatus, while it shows a lesser localization to the early endosomes in HeLa cells (Figure 2). This is generally consistent with previous findings [8,13,14,15]. While silencing of TM9SF2 did not affect either morphology or distribution of most organelles (including early endosome, late endosome, and lysosome, etc., Appendix A), TM9SF2 deficiency induced a severe Golgi fragmentation as visualized by immunostaining of trans-Golgi network marker Golgi-97. The clustered Golgi stacks that follow a juxta- or perinuclear distribution become individually dispersed throughout the cytoplasm in response to TM9SF2 knockdown (Figure 3A).

It is established that aberrant cholesterol metabolism and trafficking usually induce fragmentation of the Golgi apparatus [17]. Therefore, we sought to investigate whether cholesterol homeostasis was affected in the context of TM9SF2-knockdown cells. Intracellular cholesterol is mostly stored in lipid droplets. We detected the lipid droplets using Nile Red staining. As shown in Figure 3B, we observed a significant accumulation of lipid droplets within the cytosolic compartment of TM9SF2-knockdown cells as compared with its control group. We speculated that silencing TM9SF2 impairs cholesterol intracellular transport and promotes its esterification and storage in lipid droplets. The aberrant distribution and accumulation of cholesterol ultimately leads to the fragmentation of the Golgi complex in TM9SF2-deficient cells.

### 2.3. Pharmacological Intervention of Cholesterol Accumulation Restores Golgi Integrity and Ricin-Induced Cytotoxicity in TM9SF2 Knockdown Settings

To further substantiate the interplay between cholesterol homeostasis, Golgi morphology, and ricin tolerance, we, respectively, implemented two chemical inhibitors (A939572 and avasimibe). A939572 is an inhibitor of stearoyl-Coenzyme A desaturase 1 (SCD1). Avasimibe is an acyl-CoA:cholesterol acyltransferase 1 (ACAT1) inhibitor. Both of them are effective in preventing cholesterol accumulation [18]. As expected, both A939572 and avasimibe sufficiently reversed the cholesterol accumulation in TM9SF2-silenced cells (Figure 4A). Meanwhile, pharmacological treatment with these two chemicals indeed showed a significant reversion of Golgi fragmentation induced by TM9SF2 silence (Figure 4B). In the context of ricin stimulation, both A939572 and avasimibe substantially reversed the protection phenotype conferred by TM9SF2 knockdown and restored ricin-induced cytotoxicity in TM9SF2-deficient settings (CC_50_: siNC = 0.09 ng/mL; siTM9SF2 + DMSO = 0.23 ng/mL; siTM9SF2 + A939572 = 0.09 ng/mL; siTM9SF2 + avasimibe = 0.10 ng/mL) (Figure 4C,D). These results clearly supported that TM9SF2 mitigates ricin-induced cytotoxicity by affecting cholesterol translocation and Golgi integrity.

## 3. Discussion

TM9SF2 has been repeatedly identified as a susceptible host factor for several bacterial toxins, including Shiga toxins (Stx1 and Stx2) [8,14,19] and TcdA^1–1874^ [20]. In this study, we validated TM9SF2 as a critical host factor for ricin-induced cytotoxicity. We found that genetic interference of TM9SF2 significantly increases the cellular resistance to ricin exposure. We demonstrated that TM9SF2 knockdown results in intracellular cholesterol accumulation and Golgi fragmentation, which, in turn, disrupts the retrograde transport of ricin and thereby attenuates ricin-induced cytotoxicity. We further demonstrated that pharmacological intervention of cholesterol accumulation largely restores the integrity of the Golgi apparatus and reverses the ricin-protection phenotype induced by TM9SF2 knockdown.

Golgi morphology and integrity have been linked to ricin cytotoxicity in a few previous studies. For instance, it has been reported that brefeldin A reversibly disrupts the intracellular Golgi network and protects from ricin-induced cellular toxicity [21]. Benzyl alcohol was described as being able to induce reversible dispersion of the Golgi apparatus, thereby inhibiting transport between endosomes and the trans-Golgi network, which, in turn, reduces the toxicity of Shiga toxin and ricin [22].

Mounting evidence has shown that aberrant metabolism and disposition of cholesterol leads to structural disruption and functional impairment of the Golgi apparatus [23]. For example, inhibition of OSBP, a signaling protein at the ER–Golgi contact site, reduces cholesterol transfer from the ER to the trans-Golgi network (TGN) and impairs the structural integrity of the Golgi apparatus [24,25,26]. Similarly, the deficiency of TMED2/10 affects cholesterol flux from the ER to the Golgi and induces partial fragmentation of the Golgi apparatus [18]. In this manuscript, we revealed that there is a significant increase in the number and size of lipid droplets in cells subjected to TM9SF2 knockdown. As lipid droplets serve as major sites for cholesterol storage, these results suggested that TM9SF2 knockdown impairs cholesterol translocation and results in aberrant cholesterol accumulation. The accumulation of cholesterol subsequently leads to severe fragmentation of the Golgi complex in TM9SF2 knockdown cells.

## 4. Conclusions

Overall, we verified TM9SF2 as a critical host factor for ricin-induced cytotoxicity and demonstrated that TM9SF2 regulates ricin-induced cytotoxicity by affecting intracellular cholesterol trafficking and Golgi integrity. These findings provide novel insights into the pathogenic mechanisms of type II ribosomal inactivating proteins, including ricin.

## 5. Materials and Methods

### 5.1. Antibodies, Reagents, and Cell Lines

Antibodies for the following antigens were purchased from commercial vendors: TM9SF2, RAB7, and LAMP1 were purchased from Abcam, Cambridge, UK. Golgin-97 and EEA1 were obtained from Cell Signaling Technology (CST, Danvers, MA, USA). FLAG-tag and Tubulin were purchased from Sigma, St. Louis, MO, USA. The reagents were sourced as follows: Stx1 and Stx2 were purchased from Nanjing Mighty Biotechnology Co., Ltd. (Nanjing, China). Ricin was obtained from the State Key Laboratory of Toxicology and Medical Countermeasures at Beijing Institute of Pharmacology and Toxicology, (Beijing, China). The cell lines were sourced as follows: 5637 cell lines were purchased from Pricella, Wuhan, China and HeLa cell lines were obtained from the State Key Laboratory of Toxicology and Medical Countermeasures at the Beijing Institute of Pharmacology and Toxicology.

### 5.2. Cell Culture Conditions

HeLa cells were cultured in Dulbecco’s Modified Eagle’s Medium (DMEM) (M&C Gene Technology, Beijing, China) containing 10% fetal bovine serum (FBS, Gibco, Waltham, MA, USA) and 100 U/mL penicillin. All the cells were maintained at 37 °C under 5% CO_2_.

### 5.3. cDNA Constructs

The cDNA of TM9SF2 (NCBI RefSeq: NM_004800.3) was cloned from HeLa cells, and the amplification primers are as follows: 5′TAAGCTTGGTACCGAGCTCGGATCCATGAGCGCGAGGCTGCCGGTGTTGT-3′ (sense) and 5′-ACTGTGCTGGATATCTGCAGAATTCGTCAACCTTCACCACACTGTATATT3′ (antisense). Full-length TM9SF2 was cloned into pcDNA3.1, with a triple FLAG tag (our laboratory’s collection).

### 5.4. Transient Transfection

For gene knockdown experiments, cells were seeded into 6-well plates and incubated for 12 h prior to transient transfection with 100 nM siRNA using Lipofectamine™ RNAiMAX (Invitrogen, Carlsbad, CA, USA) according to the manufacturer’s instructions, with a standardized 36-h transfection period maintained for all experiments. The siRNA sequences are listed in Table 1. For the exogenous plasmid transfection experiments, HeLa cells were seeded in 24-well plates. Transfection was initiated after the cells adhered and the cell density reached approximately 30%. Cells were transfected with 500 ng of plasmid solution using Lipofectamine 3000 reagent (Invitrogen, Carlsbad, CA, USA) according to the manufacturer’s instructions.

### 5.5. Cell Viability Assay

HeLa cells from each group were seeded in 96-well plates. After the cells were cultured for 6 h, ricin was added at an initial concentration of 25 ng·mL⁻^1^ and diluted twofold with serum-free DMEM medium to create 13 concentration gradients. Three replicates were performed for each concentration (*n* = 3). After 48 h of ricin treatment, 10 µL of CCK-8 reagent (YEASEN, Shanghai, China) was added to each well. The plates were incubated at 37 °C for 2 h, and absorbance was measured at 450 nm using a microplate reader.

### 5.6. Crystal Violet Staining Assay

The cells from each group were passaged in 24-well plates. After the cells adhered, ricin was added at concentrations of 0.1 and 0 ng·mL⁻^1^ in serum-free DMEM medium. Each concentration was set with three replicates (*n* = 3). After 48 h of ricin treatment, the medium was removed, and an appropriate amount of methanol was added to fix the cells for 5 min. Methanol was then discarded, and crystal violet staining solution was added to cover the cells for 5 min. The cells were rinsed with distilled water, allowed to dry, and observed and photographed under a microscope.

### 5.7. qRT-PCR Assay

Total cellular RNA was extracted using the Total RNA Extraction Kit (TIANGEN, Beijing, China), quantified, and subjected to reverse transcription using the PrimeScript™ RT reagent Kit (TaKaRa, Kyoto, Japan). Quantitative reverse transcription PCR (RT-qPCR) was conducted using SYBR Green Master Mix (Thermo, Waltham, MA, USA). The relative changes in gene expression were calculated using the ΔΔCt method. Glyceraldehyde 3-phosphate dehydrogenase (GAPDH) was used as a housekeeping gene to normalize the relative mRNA levels. The primer sequences for RT-qPCR are listed in Table 2.

### 5.8. Western Blot

The cells were lysed in RIPA buffer (Beyotime, Shanghai, China). Subsequently, cell lysates were collected and protein concentrations were determined using a BCA Protein Assay Kit (Vazyme, Nanjing, China). After heat denaturation, the proteins were separated on SDS-PAGE gels and transferred to polyvinylidene fluoride (PVDF) membranes (PVH00010, Millipore, Burlington, MA, USA). The membranes were blocked for 1 h in TBST containing 5% skim milk at room temperature, then incubated with primary antibodies overnight at 4 °C. The membranes were then incubated with HRP-labeled secondary antibodies for 1 h at room temperature. Finally, membranes were developed using an ECL chemiluminescent reagent kit (Absin, Shanghai, China). The membranes were washed three times with 1% TBST after each.

### 5.9. Immunofluorescence

After transfection, HeLa cells were fixed with 4% paraformaldehyde for 20 min, permeabilized with 2% Triton-100 for 5 min, and blocked with 10% FBS at room temperature for 1 h. The cells were thoroughly washed with PBS and incubated with primary antibodies overnight at 4 °C. Incubation with both primary and secondary antibodies was performed in the same buffer. The secondary antibody was incubated at room temperature for 1 h, and the cells were treated with mounting medium containing DAPI. Finally, images were acquired using a confocal microscope with a 63× objective lens or z-scan, and the images were processed using ImageJ 1.39u.

### 5.10. Nile Red Staining Assay

HeLa cells were fixed with 4% paraformaldehyde for 20 min and washed three times with PBS. Nile Red (Abmole, Shanghai, China) was added, and the cells were incubated at room temperature for 15 min, followed by thorough washing with PBS. The cells were then treated with mounting medium containing DAPI. Images were acquired using an immunofluorescence microscope.

### 5.11. Statistical Analysis

GraphPad Prism 8.0 software (GraphPad Software, Inc., La Jolla, CA, USA) was used for the data analysis. Comparisons between two groups were performed using Student’s *t*-tests. Measurement data are expressed as mean ± SD (standard deviation), and *p*-values were utilized to assess the statistical significance of differences among the tested groups, with a threshold of *p* < 0.05 indicating statistical significance.

## Figures and Tables

**Figure 1 toxins-17-00218-f001:**
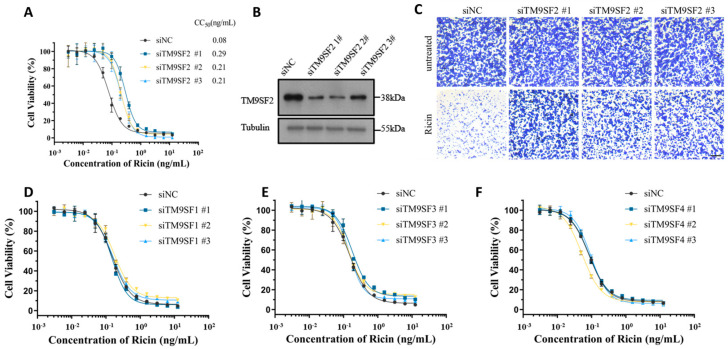
TM9SF2 is non-redundantly required for ricin-induced cytotoxicity within the TM9SF family. (**A**) TM9SF2 knockdown cells treated with three different siRNA sequences and subsequently exposed to ricin for 48 h showed significantly increased resistance compared to negative control siRNA (siNC) cells (CC_50_: siNC = 0.08 ng/mL; siTM9SF2 #1 = 0.29 ng/mL; #2 = 0.21 ng/mL; #3 = 0.21 ng/mL). Error bars (*n* = 3) indicate the mean ± SD. (**B**) Western blot analysis of TM9SF2 expression in panel A was performed using rabbit anti-TM9SF2 (1:1000), with α-Tubulin (mouse monoclonal, 1:5000) as the loading control. (**C**) siNC cells and siTM9SF2 cells were post-treated with ricin (48 h) or left untreated following siRNA pretreatment, then fixed and stained with crystal violet. Scale bar, 500 μm. (**D**–**F**) The sensitivity of siTM9SF1 (**D**), siTM9SF3 (**E**), and siTM9SF4 (**F**) cells to ricin was assessed by MTT assay following ricin exposure (48 h). Error bars (*n* = 3) indicate mean ± SD.

**Figure 2 toxins-17-00218-f002:**
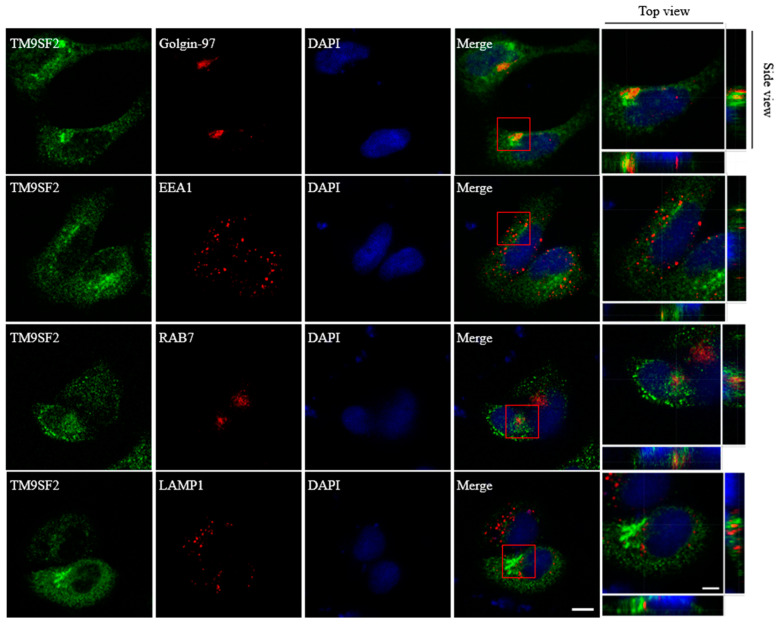
Subcellular localization of TM9SF2. Immunofluorescence analysis of HeLa cells 36 h post-transfection with exogenous TM9SF2 revealed predominant colocalization with the Golgi marker Golgin-97, as observed in the z-scan by confocal microscopy. EEA1 (in red) represents early endosomes, RAB7 (in red) represents late endosomes, LAMP1 (in red) represents lysosomes, and Golgin-97 (in red) represents the trans-Golgi network, DAPI (in blue) represents the nucleus. TM9SF2 was detected using a FLAG-tag antibody (in green), scale bar, 10 μm, 5 μm. The red box represents that the single Z-plane analysis (0.5 μm steps) confirmed the precise colocalization between X and Y within various organelle regions.

**Figure 3 toxins-17-00218-f003:**
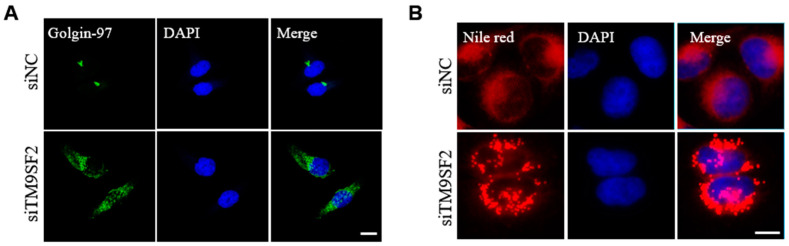
TM9SF2 silencing affects the morphology of the Golgi apparatus and cholesterol homeostasis. (**A**) Immunofluorescence analysis showed that the distribution of the Golgin-97 protein undergoes significant changes, and the Golgi apparatus becomes fragmented in siTM9SF2 cells. Golgin-97 (in green) represents the trans-Golgi network, scale bar, 20 μm. (**B**) siNC or silenced cells were fixed and stained with Nile red (in red) for 15 min at room temperature, DAPI (in blue) represents nucleus, scale bar, 10 μm.

**Figure 4 toxins-17-00218-f004:**
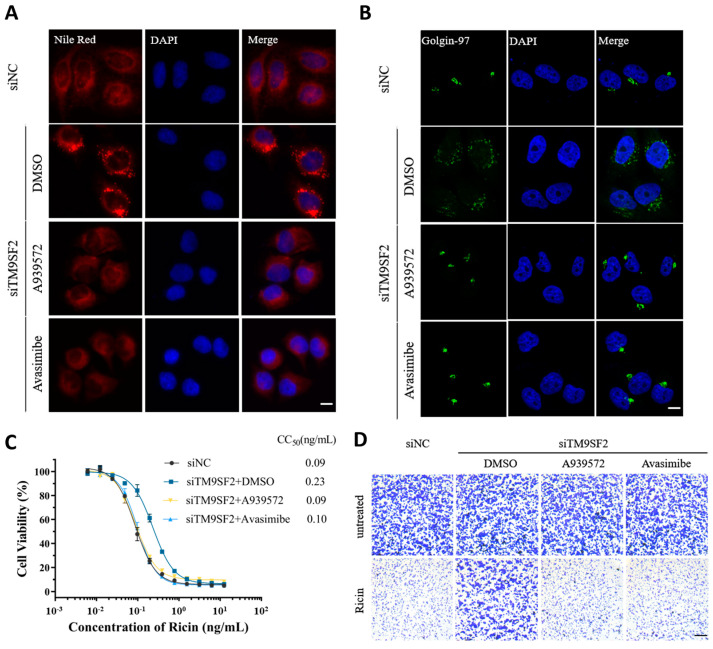
Cholesterol metabolism-related inhibitors restore Golgi structural integrity and Ricin cytotoxicity in siTM9SF2 cells. (**A**) siNC or siTM9SF2 cells were treated with either A939572 (10 µM) or avasimibe (5 µM) or not (DMSO) 24 h before fixation and were stained with Nile red (in red). (**B**) Cells in (**A**) were fixed and stained with Golgin-97 (in green) markers. DAPI (in blue) was used to stain the nucleus. All scale bars are 20 μm. (**C**) siNC or siTM9SF2 cells were pre-treated with A939572 or avasimibe or not (DMSO) for 24 h before seeded into 96-well plates and exposed to ricin for 48 h (CC_50_: siNC = 0.09 ng/mL; siTM9SF2 + DMSO = 0.23 ng/mL; siTM9SF2 + A939572 = 0.09 ng/mL; siTM9SF2 + avasimibe = 0.10 ng/mL). Error bars (*n* = 3) indicate mean ± SD. (**D**) After siNC cells and siTM9SF2 cells were pre-treated with A939572 or avasimibe or not (DMSO), the indicated cells exposed to ricin or no treatment were fixed and then stained with crystal violet, scale bar, 500 μm.

**Table 1 toxins-17-00218-t001:** siRNA Sequences.

	Sequences (5′ → 3′)
siRNA-NC	GGAATCTCAYYCGATGATGCATCA
siRNA-TM9SF2 #1	CCTTCACCATATAAGTTTACGTTTA
siRNA-TM9SF2 #2	GTGTTACGATGTTGAAGAT
siRNA-TM9SF2 #3	GCAGTACACTACTTCTTTT

**Table 2 toxins-17-00218-t002:** Primers used for qPCR.

	Forward Primers (5′-3′)	Reverse Primers (5′-3′)
TM9SF2	ATGGGCGTCTAGATGGGACT	CCTGGGCATCTTCCGTAGAG
GAPDH	ATTCCATGGCACCGTCAAGG	TGGACTCCACGACGTACTCA

## Data Availability

The original contributions presented in this study are included in the article and Appendix A. Further inquiries can be directed to the corresponding authors.

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
