# Peer review of "TM9SF2 Maintains Golgi Integrity and Regulates Ricin-Induced Cytotoxicity"

_toxins, 2025, doi:10.3390/toxins17050218_

Round 1

Reviewer 1 Report

Comments and Suggestions for Authors

The manuscript titled “TM9SF2 maintains Golgi integrity and regulates ricin-induced cytotoxicity” is well written and presents a clear experimental framework. The data are logically organized and generally well presented with appropriate statistical analysis. The findings provide novel insights into the role of TM9SF2 in regulating intracellular cholesterol trafficking and Golgi integrity, linking these to the mechanism of ricin-induced cytotoxicity.

However, several concerns, especially related to microscopy interpretation and figure clarity, need to be addressed to further strengthen the manuscript.

Major Revisions

  1. Timing of Fixation for Microscopy (Section 2.2, Figure 2 and Figure S2):
    The manuscript discusses the subcellular localization of TM9SF2, noting colocalization with Golgin-97 and early endosomes. However, it lacks critical information about the time of fixation following transfection or treatment. Since endosome morphology and localization are highly time-dependent, this omission limits interpretability. Please specify the time points (in hours or minutes) post-transfection or post-treatment in both the main text and figure captions.
  2. Unusual Early Endosome (EEA1) Morphology and Positioning (Figure 2 and Figure S2):
    The observed EEA1-positive early endosomes are unusually enlarged and perinuclear. Given that early endosomes are generally expected to localize closer to the plasma membrane as initial sorting stations, this should be addressed. Are the images showing aberrant EE morphology? Could this be due to overexpression artifacts, TM9SF2 modulation, or technical limitations? Please discuss these observations and, if possible, provide higher-resolution or representative images across multiple Z-planes to clarify.
  3. Lack of Z-Stack Views in Figures 2, 3, and 4:
    As the manuscript draws strong conclusions from immunofluorescence colocalization, it is crucial to show Z-stack reconstructions or orthogonal views (XY/XZ/YZ) to confirm colocalization across 3D space, not just in 2D projections. Please include representative Z-stack or orthogonal views to validate these key findings.

Minor Revisions

  1. Western Blot Clarity and Caption (Figure 1A):
    The Western blot data included in Figure 1A are difficult to read and lack sufficient annotation in the figure caption. Consider increasing the resolution and labeling bands clearly, or alternatively separating the blot into a new panel (e.g., Figure 1B), with full description of molecular weights, antibody used, and loading control.
  2. Unclear Abbreviation "siNC" and Typo ("siNC cell"):
    The abbreviation "siNC" is used frequently but not explicitly defined in the main text. Please define it upon first mention (e.g., "siNC: negative control siRNA"). Also, in the sentence “siNC cell,” the singular form is incorrect. It should read “siNC cells.”
  3. Consistency of Terminology – “Golgi” vs “golgi”:
    Throughout the manuscript, there are inconsistencies in capitalization (e.g., “golgi” instead of “Golgi”). Please ensure consistent usage of “Golgi” when referring to the organelle.
  4. Typographical and Grammar Errors:
    • Page 2, line 13: "impedes retrograde transport process of ricin" → should read "impedes the retrograde transport of ricin."
    • Page 3, line 86: "become dispersed singly" → awkward phrasing; consider "become individually dispersed."
    • Page 4, line 112: "interplay across cholesterol homeostasis" → revise to "interplay between cholesterol homeostasis..."

Reviewer 2 Report

Comments and Suggestions for Authors

Authors present a novel molecular mechanism on how a member of the transmembrane 9 super family (TM9SF2) mediates the toxicity of ricin in Hela cells. They verified that TM9SF2 regulates ricin-induced cytotoxicity by affecting intracellular cholesterol trafficking and Golgi integrity. This is novel and supports our knowledge on ribosome-inactivating toxins.

Unfortunately, the manuscript does not meet the expectations of a first-class paper. Only in Material and Methods does the reader learn where the TM9SF2 proteins for the experiments come from. In the Introduction, values for the lethal dosis of ricin are presented, without saying for whom they are valid (humans?). Legend to Fig. 4 is unclear. The acronym Ctl for control is not shown in the figures (DMSO?).

The manuscript needs careful grammatical and spelling check;

  • line 27 in others: all species names in italics
  • line 47: increases
  • line 72: three sequences
  • line 81: in agreement with; insert spaces before brackets
  • line 88 and others: Golgi in uppercase letters
  • line 114 and others: give names of chemicals in lowercase letters
  • line 126: the sentence is not complete
  • line 138: increases
  • line 154: explain the acronym TGN
  • line 172: tubulin?
  • line 180 and others: give all subheadings in lowercase letters
  •  line 202: sequences
  • add "merged" in last column of Fig. S2
  • References must be adapted to MDPI Authors' Instructions

Reviewer 3 Report

Comments and Suggestions for Authors

The manuscript deals with the elucidation of the mechanism of action of ricin, that the authors show to be mediated by TM9SF2.

The demonstration was achieved using gene knockdown and pharmacological inhibition of cholesterol accumulation. Although reported data are not abundant, they are sufficient to support conclusions.

Nevertheless, some improvements are required, including an overall revision of the English Language and the addition of further information (see the notes in the pdf for further details).

Comments on the Quality of English Language

I recommend an overall revision of the English Language. Among various points, please consider alternative lines 39-40 ( studies have identified TM9SF2 as one of the important host factors), line 41 ("it has been reported" probably would be more appropriate), and similar improvements throughout the text.

Reviewer 4 Report

Comments and Suggestions for Authors

Dear Editor, Dear Authors

I was invited to evaluate the manuscript « TM9SF2 maintains Golgi integrity and regulates ricin-induced cytotoxicity »

TM9SF2 has been frequently identified as one of the important host factors proteins involved on the effects of lethal pathogens and toxins. The authors investigated the role of this protein in the cytotoxicity of ricin, well-know ribotoxin. Data provided show that : i) TM9SF2 displays non-redundant requirement for ricin-induced cytotoxicity within the nonaspanin family,and that  ii) genetic interfering of TM9SF2 substantially affects/remodels intracellular cholesterol trafficking, which results in abnormal cholesterol accumulation in Golgi compartments and causes severe Golgi fragmentation. The resulting disruption of Golgi integrity and network inhibits the retrograde transport process of ricin and reducing also ricin-induced cytotoxicity. Nicely, the authors showed that pharmacological manipulation of cholesterol metabolism (e.g. by using A939572 and Avasimibe etc), which well restores the integrity the Golgi apparatus, reversed the ricin-resistant phenotype induced by TM9SF2 knockdown.

Overall, I found the data convincing and the manuscript clear.

Please find below some comments.

1- Fig 1 ad related text : please calculated and provide CC50 values of ricin in each case. Same for Fig 4.

2- since siRNA silencing of TM9SF2 results in Golgi defects, this can also lead to toxic effect of the siRNA without ricin. Please check it.

3- It would be important, to further confirm the hypothesis of the authors, to check if siRNA silencing of TM9SF2 will affect or not the toxicity of other ribotoxins that do not need golgi apparatus such as the mycotoxin deoxynivalenol causing ribotoxic effect by targetting ribosomes directly.

regards

Round 2

Reviewer 1 Report

Comments and Suggestions for Authors

The authors responded to all comments and suggestions. This version seems ready for publication.

Reviewer 4 Report

Comments and Suggestions for Authors

Dear Editor,

The authors addressed all my concerns

regards